# ROBUSTNESS OVER TIME: UNDERSTANDING ADVERSARIAL EXAMPLES' EFFECTIVENESS ON LONGITUDINAL VERSIONS OF LARGE LANGUAGE MODELS

## ABSTRACT

Large Language Models (LLMs) have led to significant improvements in many tasks across various domains, such as code interpretation, response generation, and ambiguity handling. These LLMs, however, when upgrading, primarily prioritize enhancing user experience while neglecting security, privacy, and safety implications. Consequently, unintended vulnerabilities or biases can be introduced. Previous studies have predominantly focused on specific versions of the models and disregard the potential emergence of new attack vectors targeting the updated versions. Through the lens of adversarial examples within the in-context learning framework, this longitudinal study addresses this gap by conducting a comprehensive assessment of the robustness of successive versions of LLMs, i.e., GPT-3.5 and LLaMA. We conduct extensive experiments to analyze and understand the impact of the robustness in two distinct learning categories: zero-shot learning and few-shot learning. Our findings indicate that, compared to earlier versions of LLMs, the updated versions do not exhibit the anticipated level of robustness against adversarial attacks. We hope that our study can lead to a more refined assessment of the robustness of LLMs over time and provide valuable insights into these models for both developers and users.

## 1 INTRODUCTION

Large Language Models (LLMs), such as OpenAI ChatGPT (**?**), Meta LLaMA (Touvron et al., 2023b), have demonstrated remarkable capabilities in many Natural Language Processing (NLP) tasks, including language translation (Jiao et al., 2023), text classification (Sun et al., 2023), and creative writing (He et al., 2023; Ji et al., 2023). Despite their impressive performance, these models also present certain risks. For instance, these LLMs are trained on vast amounts of internet data, which may contain biases, misinformation, and offensive content (Turpin et al., 2023; Bian et al., 2023; Ouyang et al., 2022). Consequently, the outputs generated by these models can perpetuate harmful stereotypes (Liang et al., 2022; Abid et al., 2021), disseminate false information (Azaria & Mitchell, 2023; Manakul et al., 2023; Pan et al., 2023; Hanley & Durumeric, 2023), or produce inappropriate and offensive content (Kang et al., 2023). Furthermore, previous studies have shown that LLMs are sensitive to changes in input queries, including both unintentional errors by legitimate users and intentional modifications by potential attackers (Min et al., 2022; Zhu et al., 2023b).

In response, frequent updates have been made to improve LLMs by incorporating feedback and insights from users and developers (i.e., AI-human alignment). Although such updates partially mitigate known attacks or failures observed in earlier versions of GPT-3.5 (Borji, 2023; Kang et al., 2023), unintended consequences and even new vulnerabilities or biases can still be introduced. However, current research on the robustness evaluation of LLMs has focused on a *single version* of the LLM but leaves the impact of model updates unexplored.

To fill this gap, in this paper, we undertake the first comprehensive robustness evaluation of longitudinally updated LLMs. Our study is to identify and elucidate potential issues resulting from model updates. The benefits are two-fold. From the perspective of users, understanding the limitations and risks associated with model updates enables them to make informed decisions about their usage of

LLMs. From the perspective of model owners, continuous evaluation and testing facilitate iterative improvement, addressing emerging challenges and refining model behavior over time.

**Methodology.** Our primary objective is to understand the robustness of different versions of LLMs using adversarial examples within the framework of in-context learning (ICL) (Brown et al., 2020). ICL is a training-free learning framework. It involves feeding the LLM with different components: a task *description*, a *question*, and possibly some *demonstrations* consisting of task-related examples. The LLM then learns the pattern hidden from these three elements to accomplish the task. Our goal is

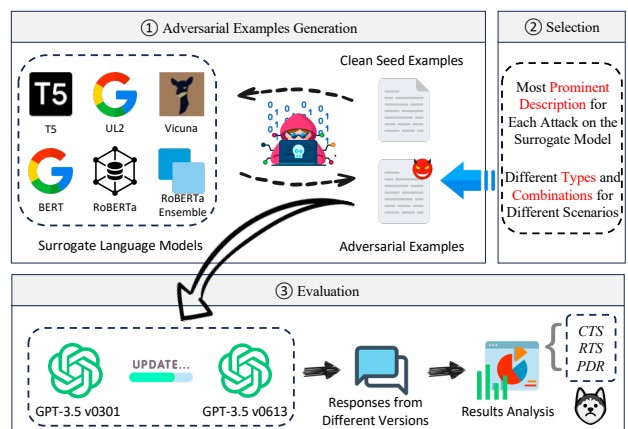

**Figure 1:** Overview of our evaluation framework on adversarial robustness of LLMs over time using adversarial examples generated from various surrogate models.

to evaluate if an LLM is robust when these elements are replaced by their adversarial versions. Figure 1 illustrates the workflow of our study. In essence, we first transfer adversarial examples generated from a surrogate language model and apply them to different versions of the target LLM. We then compare the model behaviors in the presence of adversarial examples.

To ensure a comprehensive evaluation, we adopt six different surrogate models and consider ten different settings of adversarial queries, which correspond to various combinations of adversarial or benign description, demonstration, and question (see Section 3.2 for more details). Our preliminary investigations predominantly center on different versions of GPT and LLaMA models.

In summary, we make the following key findings:

- We demonstrate that GPT-3.5 and LLaMA are both vulnerable to adversarial queries, which is persistent across different versions. For instance, on the SST2 dataset, the average result of *Robust Test Scores* (see Section 4.3) of zero-shot learning for both versions of GPT-3.5 dropped from 85.093% and 87.390% to 37.210% and 20.652%, respectively (see Figure 3).

- We simultaneously demonstrate the performance divergence among different versions of GPT-3.5 and LLaMA against benign queries. We find that the performances of the LLMs do not steadily improve with the version updating. Specifically, GPT-3.5 v0613 exhibits a discernible decline in performance in certain tasks. For instance, on the MNLI dataset, the *Clean Test Scores* (see Section 4.3) within zero-shot learning for GPT-3.5 v0613 are worse than those for GPT-3.5 v0301. Our findings highlight the complex relation between model evolution over time and task-specific robustness. This also challenges the common notion that updated models simply lead to enhancements.

**Impact.** We hope that our study can offer valuable insights for future studies on the robustness of LLMs. It is critical that developers prioritize the consideration of model robustness during the update. Proactively incorporating robustness-enhancing techniques and conducting rigorous evaluation will bolster the resilience of an LLM to both known and unknown challenges.

## 2 PRELIMINARY

### 2.1 IN-CONTEXT LEARNING

**Overview.** The core idea of In-context Learning (ICL) is to learn from analogies implied in contextual information. ICL requires a few examples to form a demonstration context and feed them

into LLM. It does not modify the parameters of an LLM and relies on the model to learn the pattern hidden in the demonstration (and accordingly generate the right output). As such, ICL dramatically reduces the computation costs for adapting LLMs to new tasks, as fine-tuning is not required. In general, there are two categories of in-context learning, namely zero-shot learning and few-shot learning. We outline their details as follows, and their examples can be found in Figure 2.

**Zero-shot Learning.** Zero-shot learning (Xian et al., 2019) is a capability enabled by LLMs, allowing them to generalize to tasks or domains they have never been explicitly trained on. As a special case of ICL, the query of zero-shot learning to an LLM (termed $Q_{zero}$ in this paper) only contains two elements: description and question (see Figure 2a), which can be formulated as follows.

$$Q_{zero} = Description + Question. \quad (1)$$

Here *Description* serves as an instructional guide for LLMs, which could provide necessary information such as task characterization and the format of the response. *Question* is an inquiry for a specific task. Zero-shot learning relies on an LLM's capability to infer both the input distribution and the desired output from a query without any demonstrations (Wei et al., 2022). Because of the simple structure of $Q_{zero}$, zero-shot learning offers great convenience and becomes the most common approach to querying LLMs in real-world scenarios. The LLM may not have necessarily been trained on some task, but it is able to do so because it has been trained to understand the structure of that task.

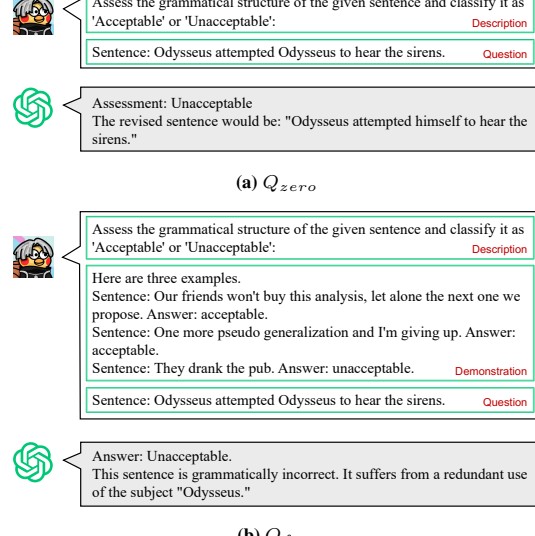

**(a)** $Q_{zero}$

**(b)** $Q_{few}$

**Figure 2:** Examples of (a) zero-shot learning and (b) few-shot learning on GPT-3.5. For zero-shot learning, the query includes only the description and the question but without any demonstrations, while few-shot learning means that the query also includes a few demonstrations.

**Few-shot Learning.** Few-shot learning (Brown et al., 2020) includes a few examples to form a demonstration context to support better the instruction/task outlined in *Description*. The examples enable LLMs to better condition the demonstration examples. Compared to zero-shot learning, few-shot learning enables LLMs to quickly adapt to new tasks by learning from an extra element, i.e., *Demonstration*. Thus, the query can be formulated as:

$$Q_{few} = Description + Demonstration + Question. \quad (2)$$

*Demonstration* typically consists of a handful of user-generated question-answer pairs. We show an example of $Q_{few}$ in Figure 2b. In general, few-shot learning can better guide the LLMs to learn a more accurate mapping between questions and desired answers. In this paper, we specifically consider 3-shot learning (i.e., three question-answer pairs in *Demonstration*).

## 2.2 Adversarial Examples

Previous studies have proposed many effective methods to generate adversarial examples against language models. However, different work considers different attacking goals of the query, i.e., description or question. For instance, Zhu et al. (2023b) proposed PromptBench to illuminate a noteworthy facet pertaining to the vulnerability of the descriptions to adversarial attacks when applied to LLMs. The notable vulnerability of descriptions to adversarial attacks, as detailed in their research, arises primarily due to their critical role in serving as a guiding framework that shapes the responses of LLMs and steers their cognitive orientation. Meanwhile, Wang et al. (2021) proposed AdvGLUE, a meticulously curated dataset comprising adversarial questions. However, to fully evaluate the robustness of the updating LLMs, we consider different types of adversarial queries, i.e., each element of a query can be clean or adversarial. The clean (adversarial) examples of each element are shown in Table 3 of Appendix.

## 3 ROBUSTNESS OVER TIME

### 3.1 OVERVIEW

Adversarial attacks remain a major threat to LLMs. They generate adversarial examples from *clean* seed queries based on a variety of adversarial attack algorithms and manipulate an LLM's behavior to elicit misleading or undesirable responses (Zhu et al., 2023b; Wang et al., 2023a;b). Moreover, recent studies have demonstrated that these adversarial examples exhibit a significant degree of transferability across different LLMs (Zhu et al., 2023b). However, mainstream LLMs are *continually updated*. As a result, the question of whether the successive iterations of these LLMs remain susceptible to previously identified adversarial strategies has not been adequately addressed, which prompts us to systematically evaluate the robustness of the latest iterations of LLMs.

In this paper, "over time" pertains to the target LLMs that undergo continuous updates under the direction of their developers. We undertake a comprehensive assessment of the robustness, focusing on GPT-3.5 and LLaMA as they are the most prominent LLMs that are subject to ongoing updates. Concretely, our analysis centers on GPT-3.5, for which two distinct versions are available: `gpt-3.5-turbo-0301` (GPT-3.5 v0301) and `gpt-3.5-turbo-0613` (GPT-3.5 v0613). For the LLaMA model, we focus on three different versions: `LLaMA`, `LLaMA2`, and `LLaMA2-chat`. Meanwhile, we consider different scales of LLaMA models. For instance, for the primary version `LLaMA`, we consider `LLaMA-7B/13B/65B`. The scales of `LLaMA2` and `LLaMA2-chat` we discuss in this paper are `7B`, `13B`, and `70B`. In accordance with the inherent structure of $Q_{zero}$ and $Q_{few}$ (see Section 2.1), we feed different type of adversarial examples into longitudinal versions of different LLMs to measure the robustness.

### 3.2 METHODOLOGY

**Outline.** As we can see in Equation 1 and Equation 2, $Q_{zero}$ and $Q_{few}$ respectively have two and three elements. Here, we classify such an LLM input as an *adversarial query* if any one of these constituent elements is generated by an adversarial attack algorithm.

**Zero-shot Learning.** We first outline the procedure for generating an adversarial query in the zero-shot learning, i.e., $Q_{zero}^{\text{adv}}$. Recall that zero-shot learning does not have any demonstration; thus, $Q_{zero}$ encompasses two elements: the description and the question. To simplify our description, we can replace description or question with adversarial (A) examples or clean (C) examples. For instance, $Q_{zero}^{\text{AC}}$ consists of an adversarial description and a clean question. In turn, we could generate the adversarial queries $Q_{zero}^{\text{adv}}$ under zero-shot learning as follows:

$$Q_{zero}^{\text{adv}} := \{Q_{zero}^{\text{AC}}, Q_{zero}^{\text{CA}}, Q_{zero}^{\text{AA}}\}.$$

**Few-shot Learning.** Similarly to the procedure employed for generating $Q_{zero}^{\text{adv}}$, we extend our approach to encompass the creation of adversarial queries in few-shot learning, i.e., $Q_{few}^{\text{adv}}$. For instance, $Q_{few}^{\text{AAC}}$ consists of an adversarial description, an adversarial demonstration, and a clean question. Given $Q_{few}$ that encompasses three distinct elements, we could generate the adversarial queries as follows:

$$Q_{few}^{\text{adv}} := \{Q_{few}^{\text{ACC}}, Q_{few}^{\text{CAC}}, Q_{few}^{\text{CCA}}, Q_{few}^{\text{AAC}}, Q_{few}^{\text{ACA}}, Q_{few}^{\text{CAA}}, Q_{few}^{\text{AAA}}\}.$$

## 4 EXPERIMENTAL SETTINGS

### 4.1 DATASETS

**Description Datasets.** For the description dataset, we select PromptBench (Zhu et al., 2023b). In this dataset, as listed in Table 3, there are ten unique seed descriptions corresponding to each clean question dataset. Ten adversarial descriptions are generated from each seed description under different levels of adversarial attacks. In this work, we choose the most prominent adversarial description for each adversarial attack algorithm according to the attack capability of the surrogate model. Consequently, in the clean description dataset, the descriptions are meticulously chosen based on the

accuracy results derived from the surrogate models. In alignment with this selection, the corresponding adversarial descriptions constitute the ensemble of the adversarial description dataset. For more generating details, please see Section A.1.

**Question Datasets.** We select six widely used benchmark question datasets, of which five are the *clean* question dataset while the other one is the *adversarial* question dataset.

- **GLUE (Wang et al., 2019)** is a collection of resources for training, evaluating, and analyzing natural language understanding systems. To align with the adversarial dataset, we choose five datasets: SST-2 (Socher et al., 2013) (sentiment analysis), QQP (Wang et al., 2017) (duplicate sentence detection), MNLI (Williams et al., 2018) (natural language inference), QNLI (Wang et al., 2019) (natural language inference), RTE (Wang et al., 2019) (natural language inference).

- **AdvGLUE (Wang et al., 2021)** involves a meticulous process aimed at crafting challenging and deceptive examples to evaluate the robustness of language models. It covers 5 natural language understanding tasks from the GLUE tasks, namely AdvSST-2, AdvQQP, AdvMNLI, AdvQNLI, and AdvRTE. They are the adversarial version of the GLUE benchmark dataset. It considers textual adversarial attacks from different perspectives and hierarchies on 3 different levels. For more generating details, please see Section A.1.

## 4.2 Models

**Surrogate Models.** In our experimental setup, we select six different surrogate language models in total. Specifically, T5 (Raffel et al., 2020), UL2 (Tay et al., 2023), and Vicuna (Vic) are used for generating *Adversarial Description*, while BERT (Devlin et al., 2019), RoBERTa (Liu et al., 2019), and RoBERTa ensemble (Liu et al., 2019) are for *Adversarial Question*.

**Target Models.** Meanwhile, we select GPT-3.5 and LLaMA as our target model for the robustness evaluation. Specifically, we employ two versions of GPT-3.5, i.e., `gpt-3.5-turbo-0301` and `gpt-3.5-turbo-0613`. In addition, we choose ten different versions of LLaMA models, i.e., `LLaMA-7B`, `LLaMA-13B`, `LLaMA-65B`, `LLaMA-2-7B`, `LLaMA-2-7B-Chat`, `LLaMA-2-13B`, `LLaMA-2-13B-Chat`, `LLaMA-2-70B`, and `LLaMA-2-70B-Chat`. The selection of these versions allows us to observe the impact of updates and improvements in the model over time.

## 4.3 Evaluation Metrics

In this paper, we consider three evaluation metrics (*CTS*, *RTS*, and *PDR*) for measuring the performance:

- *Clean Test Score (CTS)* represents the classification accuracy when testing with clean queries (i.e., $Q_{zero}^{\text{CC}}$ or $Q_{few}^{\text{CCC}}$) of the target model. It is used to evaluate the utility of the model.

- *Robust Test Score (RTS)* measures the classification accuracy score when the target model is subjected to adversarial attacks, The *RTS* serves as a standard to assess whether the model can successfully overcome adversarial attacks.

- *Performance Drop Rate (PDR)*, which was introduced by Zhu et al. (2023b), aims to quantify the extent of performance decline caused by adversarial attacks. In general, larger *PDR* means higher attack effectiveness. *PDR* can be formulated as:

$$PDR = 1 - \frac{RTS}{CTS}.$$

## 5 Evaluation

In this section, we present the robustness evaluation on longitudinal GPT-3.5. We first evaluate the zero-shot learning results (see Section 5.1) and then the few-shot learning (see Section 5.2). For each setting, we analyze the results from two different angles: model effectiveness (using *CTS* and *RTS*) and attack effectiveness (using *PDR*), aiming to show a more comprehensive evaluation of the

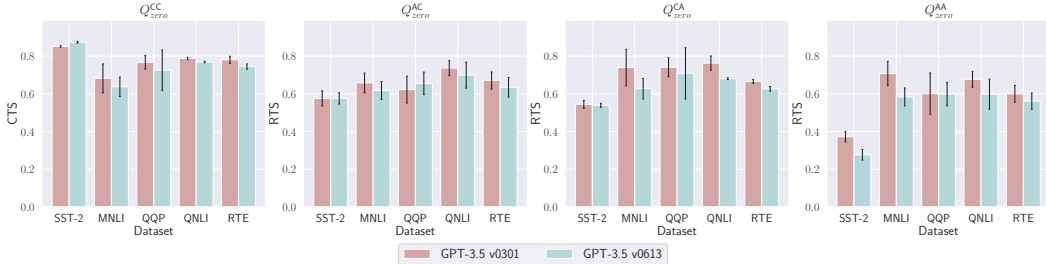

**Figure 3:** *CTS* and *RTS* on GPT-3.5 under zero-shot learning.

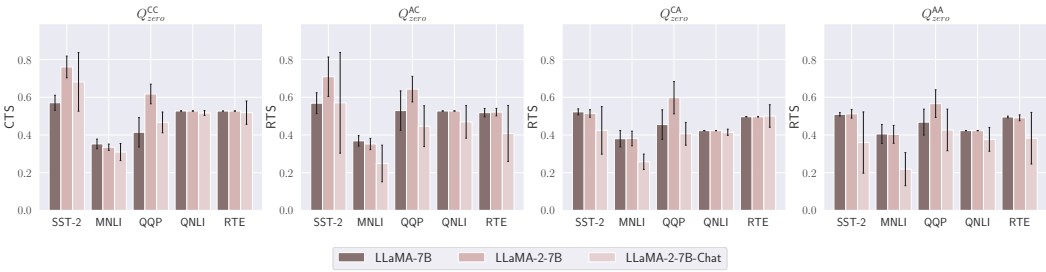

**Figure 4:** *CTS* and *RTS* on LLaMA-7B family under zero-shot learning.

longitudinal versions. Concretely, we aim to answer the following key research question: *How does the robustness of the LLMs change over time?*

**Note.** Our initial expectation was that the perturbations crafted to fool one model would not be consistently effective in deceiving an updated model (Zhao et al., 2022; Xie et al., 2019; Dong et al., 2018; Carlini & Wagner, 2017; Papernot et al., 2017). We assume the developer should have considered the adversarial examples when training or fine-tuning the model with new data. In addition, the assessment of improved robustness in an LLM necessitates a comprehensive evaluation that encompasses the aforementioned three key metrics. A conclusive determination of superiority can only be made for an updated model when it demonstrates higher *CTS* and *RTS* simultaneously, while lower *PDR*. This holistic perspective ensures a thorough examination of the model's performance across various dimensions, thereby substantiating its enhanced robustness in relation to its predecessors.

### 5.1 ZERO-SHOT LEARNING

**GPT-3.5.** We first analyze the performance of GPT-3.5 against benign queries and adversarial queries. As Figure 3 shows, the updated version of GPT-3.5 (v0613) exhibits limited advancements in terms of its overall effectiveness as compared to its earlier version (v0301). For example, in the MNLI dataset, all the *CTS* and *RTS* results from the updated version are obviously smaller than the previous version. Therefore, we could conclude that GPT-3.5 v0613 represents a retrogression of *CTS* and *RTS* than GPT-3.5 v0301.

Meanwhile, the first three rows of Table 1 show the *PDR* results of $Q_{zero}^{\text{adv}}$. Adversarial query refers to the query that contains the adversarial content in any of its two components (*description* and *demonstrations*), as defined in Equation 1. For example, the adversarial query AC means a zero-shot learning-based query that consists of an adversarial description and a clean question. Compared with GPT-3.5 v0301, besides the QQP dataset, the average values of *PDR* of GPT-3.5 v0613 model are larger than that of GPT-3.5 v0301. Upon a comparative analysis, the updated v0613 version falls short of showcasing substantial improvements in terms of effectiveness and robustness when contrasted with the v0301 iteration.

**LLaMA.** LLaMA v1 and v2 models are generative models without any instruction-tuning or RL-tuning. Thus, for these models, we choose to enlarge the logit bias for the labeled words. Figure 4 shows the *CTS* and *RTS* results of the LLaMA-7B family. The results of the LLaMA-13B and LLaMA-70B family are listed in Appendix A (see Figure 7 and Figure 8). Updated versions of the

**Table 1:** *PDR* on GPT-3.5. We highlight the larger *PDR* results. Adversarial query refers to the query that contains the adversarial content in any of its three components (*description*, *question*, and *demonstrations*), as defined in Equation 1 and Equation 2.

| ICL | Adversarial Query | SST-2 | | MNLI | | QQP | | RTE | | QNLI | |
|---|---|---|---|---|---|---|---|---|---|---|---|
| | | v0301 | v0613 | v0301 | v0613 | v0301 | v0613 | v0301 | v0613 | v0301 | v0613 |
| Zero-shot | AC | 0.324 | 0.341 | 0.036 | 0.031 | 0.188 | 0.097 | 0.064 | 0.091 | 0.141 | 0.148 |
| | CA | 0.361 | 0.383 | -0.084 | 0.015 | 0.034 | 0.023 | 0.031 | 0.115 | 0.147 | 0.159 |
| | AA | 0.563 | 0.684 | -0.037 | 0.084 | 0.217 | 0.174 | 0.140 | 0.222 | 0.232 | 0.247 |
| Few-shot | ACC | 0.013 | 0.010 | 0.040 | 0.043 | -0.002 | 0.014 | -0.012 | -0.008 | 0.024 | 0.035 |
| | CAC | -0.003 | -0.003 | 0.005 | 0.017 | 0.010 | 0.006 | 0.234 | 0.220 | 0.002 | 0.034 |
| | CCA | 0.325 | 0.363 | -0.006 | -0.097 | 0.001 | -0.023 | -0.038 | 0.040 | 0.186 | 0.078 |
| | AAC | 0.054 | 0.037 | 0.048 | 0.080 | 0.008 | 0.037 | 0.233 | 0.221 | 0.020 | 0.038 |
| | ACA | 0.358 | 0.391 | 0.020 | -0.076 | -0.000 | 0.011 | -0.045 | 0.031 | 0.215 | 0.130 |
| | CAA | 0.309 | 0.296 | 0.017 | -0.111 | 0.020 | 0.048 | -0.020 | 0.066 | 0.206 | 0.092 |
| | AAA | 0.360 | 0.354 | 0.089 | -0.007 | 0.021 | 0.057 | -0.014 | 0.085 | 0.204 | 0.141 |

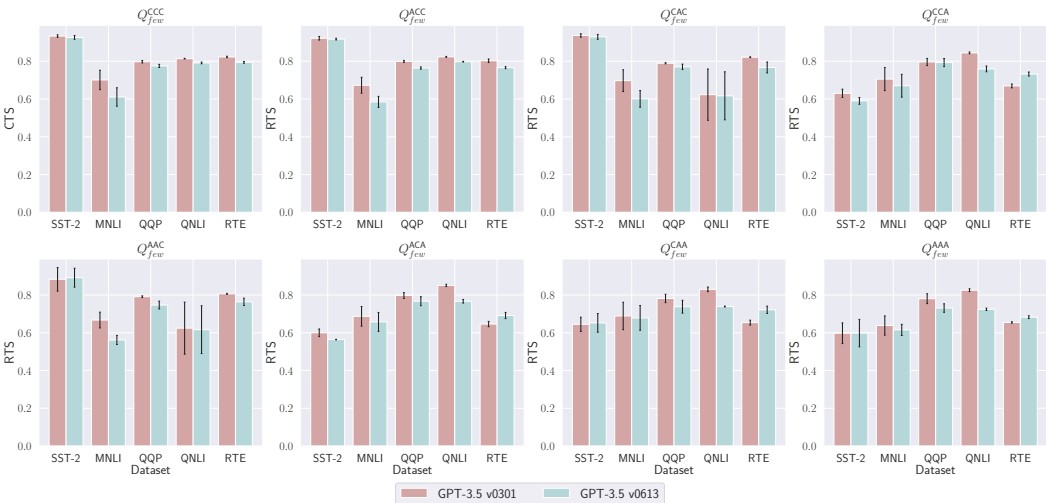

**Figure 5:** *CTS* and *RTS* on GPT-3.5 under few-shot learning.

LLaMA models exhibit improvements across several datasets, such as SST-2 and QQP. However, it is noteworthy that the v2-Chat models do not perform well as we expected. We conducted a comprehensive analysis of the results, specifically scrutinizing the output produced by these models. This examination revealed that, in numerous instances, v2-Chat models indeed encapsulate the intended meaning of the labels, albeit not in an exact match to the labels we had originally targeted. For example "not_entailment" in the Promptbench dataset is the label for the WNLI dataset, but for the tokenizer of the LLM, it will be split into five different tokens. This nuanced disparity between the model outputs and the desired labels introduces an elevated level of complexity into our evaluation process. In addition, the LLaMA models exhibit certain constraints pertaining to the generated length. These models do not possess the capability to generate the outputs according to the user-specified length requirements. Instead, they adhere to a predetermined length criterion and provide the length based on the user's need, which sometimes results in the truncation of sentences, potentially causing the omission of critically labeled words from the generated text.

For the *PDR* results, as demonstrated in the first three rows of Table 2, Table 4, and Table 5, the results from the updated models are consistently larger on average compared to the first version. As we analyzed before, we believe that the *PDR* results of v2-Chat models underscore the intricate and context-dependent nature of model behavior and robustness. The effectiveness is contingent upon diverse factors, such as generated length and label words, yielding varying results across different datasets and scenarios.

**Table 2:** *PDR* on LLaMA 7B family. Adversarial query refers to the query that contains the adversarial content in any of its three components (*description*, *question*, and *demonstrations*), as defined in Equation 1 and Equation 2.

| ICL | Adversarial Query | SST-2 | | | MNLI | | | QQP | | | RTE | | | QNLI | | |
|---|---|---|---|---|---|---|---|---|---|---|---|---|---|---|---|---|
| | | v1 | v2 | v2-Chat | v1 | v2 | v2-Chat | v1 | v2 | v2-Chat | v1 | v2 | v2-Chat | v1 | v2 | v2-Chat |
| Zero-shot | AC | 0.004 | 0.068 | 0.163 | -0.046 | -0.051 | 0.196 | -0.278 | -0.042 | 0.043 | 0.000 | 0.000 | 0.091 | 0.016 | 0.011 | 0.213 |
| | CA | 0.085 | 0.324 | 0.377 | -0.078 | -0.138 | 0.167 | -0.099 | 0.032 | 0.130 | 0.197 | 0.197 | 0.198 | 0.057 | 0.058 | 0.034 |
| | AA | 0.107 | 0.327 | 0.472 | -0.149 | -0.203 | 0.293 | -0.130 | 0.083 | 0.086 | 0.197 | 0.197 | 0.270 | 0.060 | 0.068 | 0.261 |
| Few-shot | ACC | 0.004 | -0.008 | 0.142 | 0.002 | 0.012 | 0.304 | -0.017 | -0.113 | -0.104 | 0.000 | 0.000 | 0.002 | 0.000 | 0.000 | -0.085 |
| | CAC | 0.000 | 0.000 | 0.033 | 0.000 | 0.000 | -0.145 | 0.001 | 0.000 | 0.026 | 0.000 | 0.000 | -0.126 | 0.000 | 0.000 | -0.252 |
| | CCA | 0.386 | 0.279 | 0.086 | -0.060 | -0.066 | -0.069 | 0.164 | -0.097 | 0.027 | 0.197 | 0.197 | 0.154 | 0.058 | 0.058 | -0.036 |
| | AAC | 0.004 | -0.008 | 0.198 | 0.002 | 0.011 | 0.167 | -0.018 | -0.113 | -0.019 | 0.000 | 0.000 | 0.001 | 0.000 | 0.000 | -0.274 |
| | ACA | 0.426 | 0.297 | 0.237 | -0.053 | -0.042 | 0.283 | 0.122 | -0.118 | -0.076 | 0.197 | 0.197 | 0.166 | 0.058 | 0.058 | -0.170 |
| | CAA | 0.437 | 0.373 | 0.099 | 0.016 | -0.405 | -0.145 | 0.113 | 0.086 | 0.066 | 0.197 | 0.197 | 0.084 | 0.058 | 0.058 | -0.198 |
| | AAA | 0.403 | 0.374 | 0.335 | -0.014 | -0.338 | 0.216 | 0.205 | 0.070 | 0.029 | 0.197 | 0.197 | 0.141 | 0.058 | 0.058 | -0.188 |

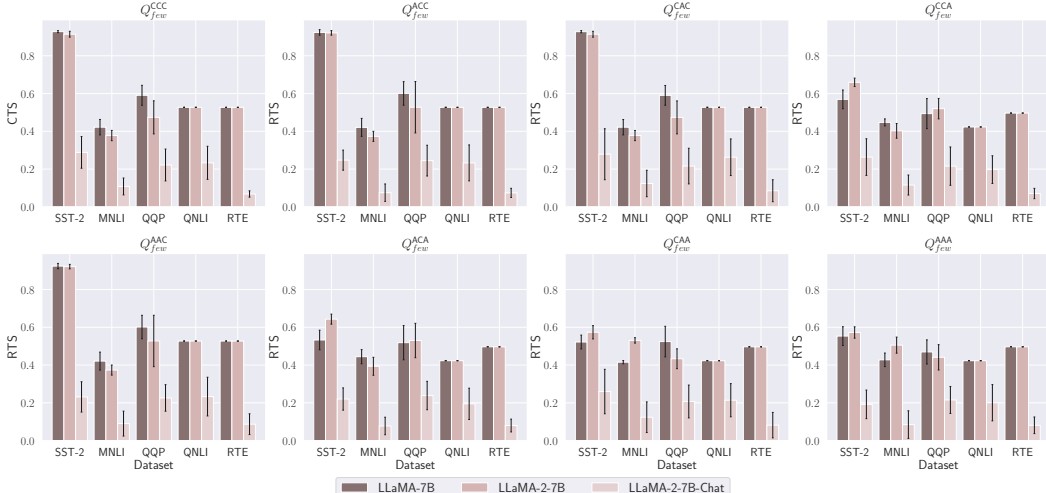

**Figure 6:** *CTS* and *RTS* on LLaMA 7B family under few-shot learning.

## 5.2 FEW-SHOT LEARNING

**GPT-3.5.** Figure 5 illustrates the *CTS* and *RTS* results of different categories of queries. Compared with zero-shot learning, the *RTS* results underscore the context-dependent nature of model behavior and the robustness. Upon a comparative analysis, the updated v0613 version falls short of showcasing substantial improvements in terms of model effectiveness when contrasted with the v0301 iteration. For instance, on the MNLI dataset, both the *CTS* and *RTS* results under each scenario for the v0613 model are discernibly lower than those of the preceding version.

As demonstrated in Table 1, the *PDR* results fluctuate from different adversarial queries. As previously indicated, we maintain the stance that these lower results alone are inadequate for rendering a conclusive judgment on the superiority of the updated version. The rationale for this perspective is discernible. Although some of the *PDR* results in the updated version are lower than before, the decreased *CTS* and *RTS* values, such as the MNLI dataset, reinforce the notion that the updated version has not markedly improved. This consistent observation underscores that specific scenarios maintain a significant degree of attack effectiveness, even in the updated version.

**LLaMA.** For the few-shot learning, Figure 6 shows the results of *CTS* and *RTS* of the LLaMA-7B family. The results of the LLaMA-13B and LLaMA-70B family are listed in Appendix A (see Figure 9 and Figure 10). Firstly, when comparing with zero-shot learning, it is evident that the inclusion of demonstrations leads to notable enhancements in numerous outcomes. Moreover, it becomes apparent that the augmentation of a model's weight count correlates positively with an increase in its overall robustness. Nevertheless, the v2-Chat variant consistently underperforms in comparison to the standard version. Similar to our observations in zero-shot learning, it is our

contention that the LLaMA models exhibit a degree of sensitivity to the labels despite the presence of demonstration. Additionally, the fixed length for generated output is a key factor contributing to diminished performance.

For the *PDR* results, the updated version of LLaMA can be resistant to adversarial attacks compared with the v1 versions. However, it is noteworthy that the *PDR* metric is a quotient derived from the division of *CTS* and *RTS*. Considering the aforementioned analysis, it becomes apparent that the smaller *PDR* values can be attributed to potential reductions in both *CTS* and *RTS*, especially for the v2-Chat versions.

### 5.3 TAKEAWAYS

In summation, the updated version of both GPT-3.5 and LLaMA fails to deliver substantial enhancements in terms of model effectiveness, as evidenced by the observed results where the results do not align with our earlier assertions. In addition, our findings further underscore the necessity of a comprehensive evaluation framework that encompasses multiple variables when comparing the longitudinal versions of LLMs.

## 6 RELATED WORK

### 6.1 ADVERSARIAL ATTACKS

We focus on adversarial attacks that manipulate legitimate inputs to mislead a trained model to produce incorrect outputs in the NLP domain (Zhang et al., 2020). These attacks commonly manipulate the input text at character-, word-, and sentence-level to attain the attack goals (i.e., targeted or untargeted attacks). Similar to adversarial attacks in computer vision domain, they can be categorized into black-box attacks (paraphrase (Iyyer et al., 2018; Ribeiro et al., 2018; Alzantot et al., 2018), text manipulation (Belinkov & Bisk, 2018; Li et al., 2019; Minervini & Riedel, 2018), etc.) and white-box attacks (FGSM (Liang et al., 2018; Samanta & Mehta, 2018), JSMA (Papernot et al., 2016), HotFlip (Ebrahimi et al., 2018), etc.). In the NLP domain, those attacks have been successfully applied to attack various applications, such as optical character recognition (Shazeer & Stern, 2018), image caption (Chen et al., 2018), visual question answering (Xu et al., 2018), etc. Our objective here is not to devise novel adversarial attacks against LLMs. Rather, we use existing methods to understand if LLMs can be challenged by carefully crafted textual adversarial examples and if/how these adversarial examples can be transferred in different versions of an LLM.

### 6.2 LLMs

Large language models (LLMs) have become a prominent area of research and application in the NLP domain, driven primarily by the transformer architecture (Vaswani et al., 2017). These models are trained on massive text data and boast a substantial number of parameters, often exceeding hundreds of billions (**?**). Notable LLMs include GPT-4 (gpt), PaLM (Chowdhery et al., 2022), LLaMA (Touvron et al., 2023a), and Alpaca (sta). As LLMs grow in size, they demonstrate emergent abilities such as enhanced language understanding (Zhu et al., 2023a), coherent text generation (Chung et al., 2023), and contextual comprehension (Zhou et al., 2023b), which are not present in smaller models. Moreover, fine-tuning techniques, such as LoRA (Hu et al., 2022), are invented to adapt the pre-trained LLMs to specific downstream tasks, allowing them to exhibit specialized behavior and produce task-specific outputs.

## 7 CONCLUSION

We conduct a comprehensive assessment of the robustness of the longitudinal versions of LLMs with a focus on GPT-3.5 and LLaMA. Our empirical results consistently demonstrate that, for both GPT-3.5 and LLaMA, the updated model does not exhibit heightened robustness against the proposed adversarial queries compared to its predecessor. Subsequent analysis reveals a prevalent trend of decreased adversarial robustness in the updated version. Our findings reinforce the importance of understanding and assessing the robustness aspect when updating LLMs, calling for enhanced focus on comprehensive evaluation and reinforcement strategies to counter evolving adversarial challenges.

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

# A   ADDITIONAL EXPERIMENTAL RESULTS

## A.1   ADVERSARIAL QUERY

**Adversarial Description.** We adopt the PromptBench dataset Zhu et al. (2023b). The approach encompasses the generation of *Adversarial Descriptions* through the employment of seven distinct adversarial attack algorithms strategically deployed at the character, word, sentence, and semantic levels:

- **Character-level** adversarial examples are constructed through various operations such as character addition, deletion, repetition, replacement, and permutation within words of the source text. For this purpose, two prominent adversarial attack methods are utilized, namely *TextBugger* Li et al. (2019) and *DeepWordBug* Gao et al. (2018).
- **Word-level** adversarial examples are designed to deceive LLMs by replacing words with synonyms or contextually similar alternatives. To achieve this, two prominent adversarial attack methods are employed, i.e., *TextFooler* Jin et al. (2020) and *BertAttack* Li et al. (2020).
- **Sentence-level** adversarial examples are the irrelevant or extraneous sentences used for attacking the LLMs Two approaches are incorporated, namely *StressTest* Naik et al. (2018) and *CheckList* Ribeiro et al. (2020).
- **Semantic-level** adversarial examples are used to simulate the linguistic behavior of individuals from diverse countries, focusing on six commonly spoken languages: Chinese, French, Arabic, Spanish, Japanese, and Korean. These prompts are subsequently translated into English, introducing linguistic nuances and variations inherent to each language.

**Adversarial Question.** In our study, we delve into the AdvGLUE Wang et al. (2021), a meticulously curated dataset. AdvGLUE undertakes a holistic approach to assessment, strategically leveraging a compendium of 14 distinct textual adversarial attack methodologies. These methods, spanning word-level, sentence-level, and meticulously handcrafted examples, are systematically employed to challenge and interrogate the resilience and adaptability of language models across various original tasks.

- **Word-level** adversarial examples are generated from *TextBugger* Li et al. (2019), *TextFooler* Jin et al. (2020), *BertAttack* Li et al. (2020), *Sememe-PSO* Wang et al. (2020), and *CompAttack* Wang et al. (2021).
- **Sentence-level** adversarial examples are generated from *T3* Wang et al. (2020), *AdvFever* Thorne & Vlachos (2019), *SCPN* Iyyer et al. (2018), *StressTest* Naik et al. (2018), and *CheckList* Ribeiro et al. (2020).
- **Human-crafted Examples** are sampling from several adversarial dataset and attacks such as *ANLI* Nie et al. (2020), *CheckList* Ribeiro et al. (2020), and *Adv SQuAD* Jia & Liang (2017).

**Table 3:** Instances of Adversarial Description and Adversarial Question on SST-2 task.

| Element | Type | Instances |
|---------|------|-----------|
| Description | Seed | Evaluate the sentiment of the given text and classify it as 'positive' or 'negative': |
| | Adversarial | Evaluate the sentiment of the given text and classify it as 'positive' or 'negative' 5yWbBXztUY: |
| Question | Seed | Some actors have so much charisma that you 'd be happy to listen to them reading the phone book. |
| | Adversarial | Some actors have so much charisma that you 'd be jovial to listen to them reading the phone book. |

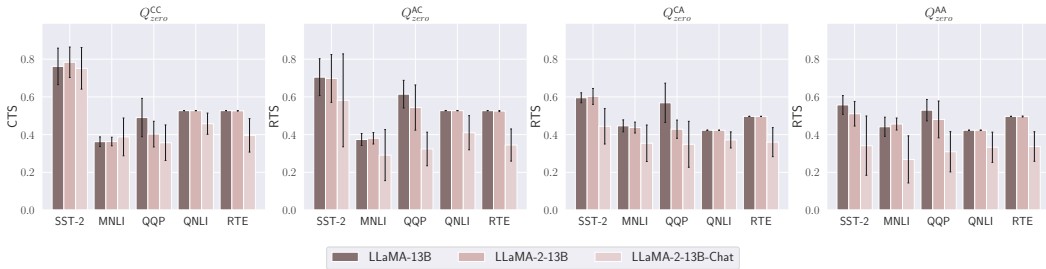

**Figure 7:** *CTS* and *RTS* on LLaMA 13B family under zero-shot learning.

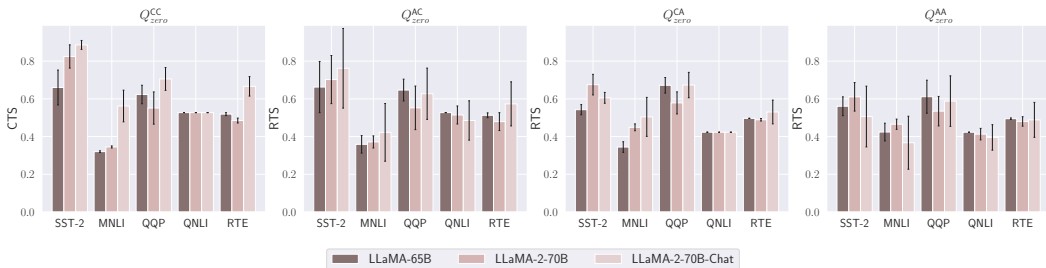

**Figure 8:** *CTS* and *RTS* on LLaMA 13B family under zero-shot learning.

## B  DISCUSSION

In this study, we do not generate adversarial examples on the *Adversarial Description* dataset and *Adversarial Question* dataset. The primary reason behind our decision lies in the considerable financial cost of generating such adversarial examples, as it requires us to query GPT-3.5 for an extensive duration, spanning several weeks. Furthermore, it is worth noting that no universally applicable standard template for the ICL exists. Certain queries may prove effective only when applied to specific datasets, and even minor modifications in the queries or choice of words can yield vastly different classification results, which increases the difficulty of the queries. Despite these limitations, our investigation reveals that the adversarial examples derived from a traditional model continue to exert a significant impact on Language Model Models (LLMs), which are in line with other previous works (Zhu et al., 2023b; Zou et al., 2023). Given that organizations such as OpenAI and Meta have not open-sourced the training datasets for their models, there exists a potential risk of inadvertently evaluating models on their training sets. While our analysis suggests a markedly low probability of our evaluation test set intersecting with the training set, accurately assessing this risk for future model iterations remains challenging. It is a generally very trending topic to construct or select non-overlap datasets for LLM evaluation nowadays Zhou et al. (2023a). In the future, we could also choose other (existing) datasets that do not overlap with the training data, e.g., the dataset with a CC BY-SA 4.0 license or constructed by ourselves. In general, we believe it is important for the model provider to use enhancing adversarial robustness methods in model upgrades. In the future, we will continue to measure the new features of LLMs after the model updates, such as browsing with Bing in GPT models.

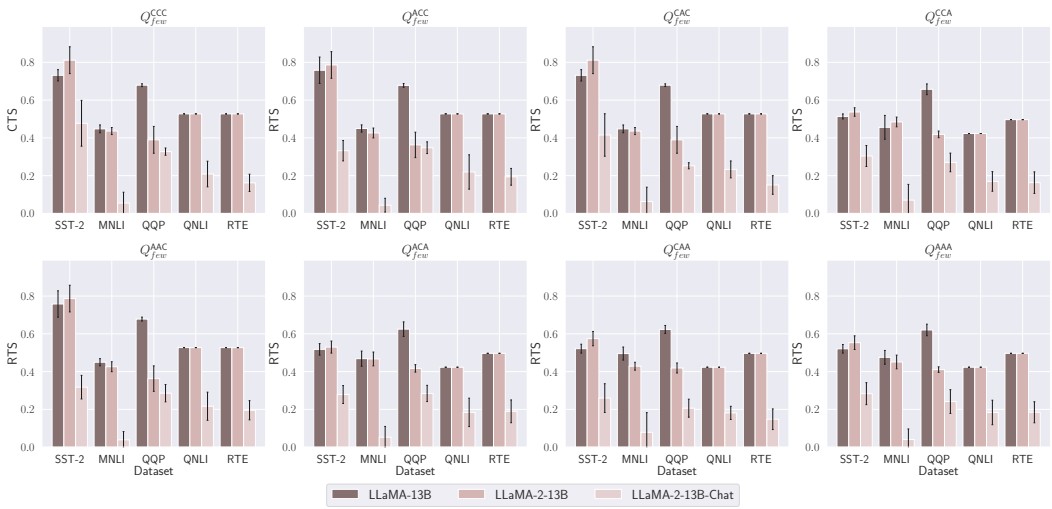

**Figure 9:** *CTS* and *RTS* on LLaMA 13B family under few-shot learning.

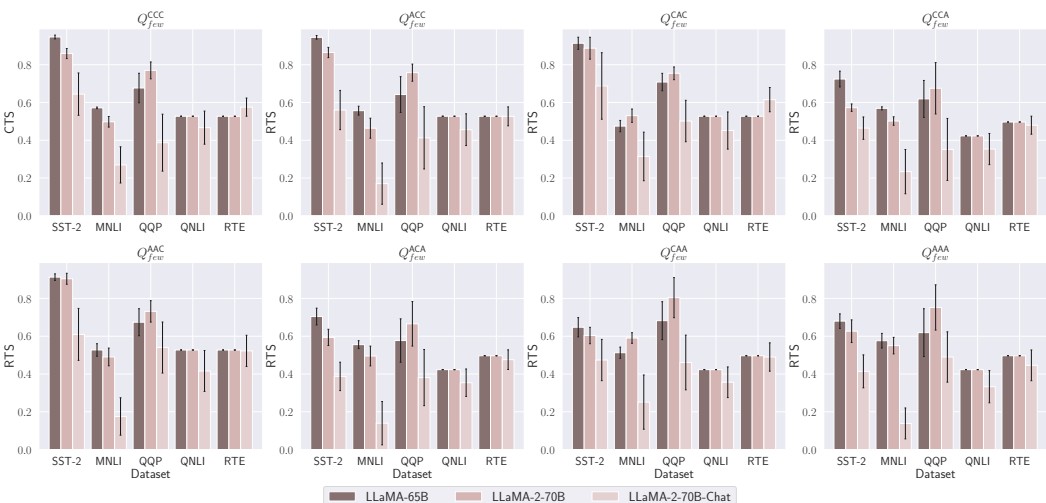

**Figure 10:** *CTS* and *RTS* on LLaMA 65B and 70B family under few-shot learning.

**Table 4:** *PDR* on LLaMA 13B family. Adversarial query refers to the query that contains the adversarial content in any of its three components (*description*, *question*, and *demonstrations*), as defined in Equation 1 and Equation 2.

| ICL | Query | SST-2 | | | MNLI | | | QQP | | | RTE | | | QNLI | | |
|---|---|---|---|---|---|---|---|---|---|---|---|---|---|---|---|---|
| | | v1 | v2 | v2-Chat | v1 | v2 | v2-Chat | v1 | v2 | v2-Chat | v1 | v2 | v2-Chat | v1 | v2 | v2-Chat |
| Zero-shot | AC | 0.075 | 0.110 | 0.226 | -0.032 | -0.048 | 0.249 | -0.253 | -0.350 | 0.093 | 0.000 | 0.000 | 0.104 | 0.000 | 0.002 | 0.130 |
| | CA | 0.218 | 0.231 | 0.409 | -0.232 | -0.204 | 0.087 | -0.159 | -0.065 | 0.024 | 0.197 | 0.197 | 0.188 | 0.058 | 0.056 | 0.091 |
| | AA | 0.268 | 0.348 | 0.546 | -0.218 | -0.256 | 0.308 | -0.080 | -0.194 | 0.134 | 0.197 | 0.197 | 0.273 | 0.058 | 0.057 | 0.149 |
| Few-shot | ACC | -0.037 | 0.031 | 0.304 | -0.005 | 0.023 | 0.204 | 0.002 | 0.068 | -0.063 | 0.000 | 0.000 | -0.052 | 0.000 | 0.000 | -0.197 |
| | CAC | 0.000 | -0.000 | 0.130 | -0.000 | 0.000 | -0.150 | 0.000 | -0.000 | 0.229 | 0.000 | 0.000 | -0.117 | 0.000 | 0.000 | 0.071 |
| | CCA | 0.298 | 0.338 | 0.363 | -0.018 | -0.110 | -0.282 | 0.032 | -0.074 | 0.175 | 0.197 | 0.197 | 0.188 | 0.058 | 0.058 | -0.006 |
| | AAC | -0.036 | 0.030 | 0.335 | -0.004 | 0.024 | 0.292 | 0.001 | 0.069 | 0.127 | 0.000 | 0.000 | -0.038 | 0.000 | 0.000 | -0.206 |
| | ACA | 0.292 | 0.347 | 0.416 | -0.047 | -0.071 | 0.038 | 0.080 | -0.071 | 0.131 | 0.197 | 0.197 | 0.118 | 0.058 | 0.058 | -0.171 |
| | CAA | 0.287 | 0.292 | 0.455 | -0.107 | 0.017 | -0.432 | 0.082 | -0.076 | 0.370 | 0.197 | 0.197 | 0.131 | 0.058 | 0.058 | 0.088 |
| | AAA | 0.288 | 0.318 | 0.406 | -0.062 | -0.034 | 0.265 | 0.086 | -0.054 | 0.262 | 0.197 | 0.197 | 0.120 | 0.058 | 0.058 | -0.140 |

**Table 5:** *PDR* on LLaMA-65B (denoted as v1), LLaMA2-70B (denoted as v2), and LLaMA2-70B-Chat (denoted as v2-Chat). Adversarial query refers to the query that contains the adversarial content in any of its three components (*description*, *question*, and *demonstrations*), as defined in Equation 1 and Equation 2.

| ICL | Query | SST-2 | | | MNLI | | | QQP | | | RTE | | | QNLI | | |
|---|---|---|---|---|---|---|---|---|---|---|---|---|---|---|---|---|
| | | v1 | v2 | v2-Chat | v1 | v2 | v2-Chat | v1 | v2 | v2-Chat | v1 | v2 | v2-Chat | v1 | v2 | v2-Chat |
| Zero-shot | AC | -0.004 | 0.149 | 0.140 | -0.121 | -0.077 | 0.249 | -0.038 | -0.002 | 0.111 | 0.000 | 0.024 | 0.079 | 0.012 | 0.012 | 0.139 |
| | CA | 0.178 | 0.181 | 0.317 | -0.076 | -0.302 | 0.103 | -0.078 | -0.049 | 0.046 | 0.197 | 0.197 | 0.197 | 0.043 | -0.009 | 0.204 |
| | AA | 0.151 | 0.259 | 0.429 | -0.324 | -0.349 | 0.346 | 0.019 | 0.030 | 0.167 | 0.197 | 0.216 | 0.249 | 0.047 | 0.010 | 0.267 |
| Few-shot | ACC | 0.002 | -0.006 | 0.131 | 0.026 | 0.068 | 0.368 | 0.051 | 0.015 | -0.066 | 0.000 | 0.000 | 0.023 | 0.000 | 0.000 | 0.085 |
| | CAC | 0.035 | -0.033 | -0.067 | 0.169 | -0.065 | -0.166 | -0.047 | 0.020 | -0.295 | 0.000 | 0.000 | 0.033 | 0.000 | 0.000 | -0.069 |
| | CCA | 0.235 | 0.334 | 0.279 | 0.005 | -0.008 | 0.132 | 0.085 | 0.123 | 0.093 | 0.197 | 0.197 | 0.243 | 0.058 | 0.058 | 0.166 |
| | AAC | 0.036 | -0.052 | 0.054 | 0.078 | 0.015 | 0.351 | 0.004 | 0.050 | -0.394 | 0.000 | 0.000 | 0.109 | 0.000 | 0.000 | 0.092 |
| | ACA | 0.256 | 0.309 | 0.399 | 0.029 | 0.005 | 0.483 | 0.147 | 0.135 | 0.017 | 0.197 | 0.197 | 0.242 | 0.058 | 0.058 | 0.173 |
| | CAA | 0.316 | 0.297 | 0.264 | 0.103 | -0.186 | 0.068 | -0.008 | -0.045 | -0.190 | 0.197 | 0.197 | 0.237 | 0.058 | 0.058 | 0.149 |
| | AAA | 0.282 | 0.271 | 0.357 | -0.008 | -0.105 | 0.487 | 0.085 | 0.023 | -0.265 | 0.197 | 0.197 | 0.287 | 0.058 | 0.058 | 0.225 |

