# OpenReview forum: "Robustness Over Time: Understanding Adversarial Examples’ Effectiveness on Longitudinal Versions of Large Language Models"
_ICLR.cc/2024/Conference — Submitted to ICLR 2024_

### Official Review · Reviewer_6TAn · 2023-10-20

**Soundness:** 2 fair
**Presentation:** 3 good
**Contribution:** 1 poor
**Rating:** 5
**Confidence:** 4

**Summary:**

This paper conducts a comprehensive experimental investigation to evaluate the robustness of updated large language models (LLMs) in comparison to their earlier versions. Utilizing established adversarial benchmarks, the research employs two distinct experimental setups: zero-shot and few-shot prediction paradigms. Contrary to expectations, the findings reveal that the newer versions of LLMs do not demonstrate a significantly enhanced level of robustness against adversarial attacks.

**Strengths:**

This paper is clearly written and straightforward to understand.

It focuses on the intriguing question of comparing the robustness between earlier and later versions of the model.

This paper offers a thorough evaluation across various scenarios, encompassing benign and adversarial descriptions, questions, and demonstrations.

**Weaknesses:**

1. My primary concern with this paper is its limited scope in terms of technical innovation. While the paper considers a range of models against existing benchmarks, albeit with some modifications and combinations, it fails to introduce new evaluation benchmarks or methodologies. Therefore, I think the paper does not meet the standards of ICLR.

2. Another issue is the selection of datasets for evaluation. The benchmarks employed are commonly used, potentially even in the training of the GPT models under scrutiny. This compromises the conclusions of the results.

3. Furthermore, the objective behind updating from GPT-3.5 v0301 to GPT-3.5 v0613 may not exclusively target robustness enhancement. Other factors such as reasoning ability, following prompts, and computational efficiency could also be taken into account. Thus, expecting substantial improvements in robustness may not be reasonable.

**Questions:**

NA

---

> ### Author Response · Authors · 2023-11-15
>
> Thank you for your elaborate review.  In the following, your comments are first stated and then followed by our point-by-point responses.
>
> >**My primary concern with this paper is its limited scope in terms of technical innovation.**
>
> We believe we introduced a new evaluation methodology, i.e.,  adversarial robustness should be evaluated `over time`. The other three reviewers highly appreciate this new angle of research. Developing new attacks/benchmarks is of general interest but would not solve (and is not relevant to) the problem that LLMs are vulnerable even to existing attacks in existing benchmarks. We also believe that uncovering the LLMs’ vulnerability to existing (simple) attacks poses more severe threats than newly designed (advanced) attacks.
>
> > **The benchmarks employed are commonly used, potentially even in the training of the GPT models under scrutiny.**
>
> This is a good point, and we will add a related discussion in the paper based on the following. In the LLM era, an LLM's training data may (partially and inadvertently) include the evaluation/test data. However, we point out that our evaluation data is not included in the training data of both GPT and LLaMA models because:
> - Regarding GPT-3.5, the GPT-3.5 training data is up to Sep 2021 according to OpenAI's website (see https://platform.openai.com/docs/models/gpt-3-5). The `AdvGLUE` dataset, however, was published in Nov 2021, and the `PromptBench` dataset was published in 2023. Therefore, It is safe to conclude that there is no possibility of training the GPT-3.5 model using these datasets.
> - Regarding LLaMA-1/2, their paper (https://arxiv.org/abs/2302.13971) states they used the GitHub dataset with `Apache`, `BSD`, and `MIT licenses`. However, the `AdvGLUE`'s license is `CC BY-SA 4.0`. Thus, Meta cannot choose `ADVGLUE` to train the model. In addition, `PromptBench` is a brand new dataset generated using different LLMs from 5 months ago, one month before the LLaMA-2 was released. Therefore, it is very unlikely for Meta to use this dataset for training the LLaMA-2 model.
> More generally, adversarial data are much less likely than clean data to be included in any natural dataset because they are specifically optimized. Our finding that LLMs are vulnerable to our adversarial data also suggests that they are not used for training.
>
> > **The objective behind updating from GPT-3.5 v0301 to GPT-3.5 v0613 may not exclusively target robustness enhancement. Thus, expecting substantial improvements in robustness may not be reasonable.**
>
> We agree that robustness is not *exclusively* important, and we do not claim that. However, the security of LLMs is an emerging research topic, and achieving adversarial robustness is a key requirement. For example, OpenAI has established the `preparedness` team to build a robust framework for monitoring, evaluation, prediction, and protection against the dangerous capabilities of frontier AI systems [4]. Following this research trend, we provide new insights that updated LLMs are vulnerable even to existing attacks.
>
>
> [4]. https://openai.com/blog/frontier-risk-and-preparedness

---

> > ### Comment · Reviewer_6TAn · 2023-11-20
> >
> > Thank the authors for the detailed reply. I have some further questions.
> >
> > > We believe we introduced a new evaluation methodology, i.e., adversarial robustness should be evaluated over time.
> >
> > I don't think introducing robustness evaluation over time is a significant novelty. I think almost all existing benchmarks include the time of a model. Furthermore, before releasing a model, all companies considered evaluating it with its prior version.  For example, the LLAMA2 paper (https://arxiv.org/pdf/2307.09288.pdf) compares LLAMA2 and LLAMA1. If they do not find any improvements, they won't even release it. Could the authors provide more justifications for why this is novel?
> >
> > > We point out that our evaluation data is not included in the training data.
> >
> > I agree with it. However, I assume the proposed benchmark will be used for future models. Recently, OpenAI introduced new models (https://platform.openai.com/docs/models/gpt-4-and-gpt-4-turbo) and the data is up to April 2023. What is the best way to compare the models over time then?
> >
> > > We agree that robustness is not exclusively important, and we do not claim that.
> >
> > In the abstract, the authors state "Our findings indicate that, compared to earlier versions of LLMs, the updated versions do not exhibit the anticipated level of robustness against adversarial attacks." I am not sure what level of improvement we should anticipate. If the model is updated for function calling (https://openai.com/blog/function-calling-and-other-api-updates), what should we expect about the robustness?
> >
> > Other than those concerns, I think the paper is generally interesting. I raise the score to 5 for now.

---

> ### Author Response · Authors · 2023-11-21
>
> Thanks for your constructive comments and suggestions; they are exceedingly helpful in improving our paper.
>
> >**I don't think introducing robustness evaluation over time is a significant novelty.**
>
> We advocate for a distinctive perspective in measuring LLM performance by examining the robustness of models updated `over time`. Note that LLMs are publicly available for millions of users. Robustness is, therefore, an important aspect for owners of LLMs since minor user input errors could result in unforeseen consequences. Unfortunately, our empirical findings reveal a prevailing oversight in addressing this concern. We acknowledge that the LLaMA framework incorporates robustness considerations within its training data. Nonetheless, our results indicate that their evaluations may overestimate the robustness of the updated model since even existing attacks considered in our work perform better on the updated model. Therefore, there remains a necessity for a holistic evaluation against adversarial attacks of the LLMs across various longitudinal versions.
>
> >**I assume the proposed benchmark will be used for future models**
>
> It is an interesting point to consider the future compatibility of our evaluation methodology. This is indeed why we focus on the methodology of  `over-time robustness evaluation` independent of any dataset. Specifically, we indeed adopt existing datasets in our work. Following the same methodology, in the future, we could also choose other (existing) datasets that do not overlap with the training data, e.g., **the dataset with a CC BY-SA 4.0 license** or **constructed by ourselves**. It is a generally very trending topic to construct or select non-overlap datasets for LLM evaluation nowadays [5, 6]. We would like to discuss this topic further in the revised version.
>
> >**I am not sure what level of improvement we should anticipate. If the model is updated for function calling, what should we expect about the robustness?**
>
> We thank the reviewer for pointing out the function calling service. This service provides a promising angle to incorporate our new finding that a new version of the model may not be more robust than its older counterpart. Specifically, based on our evaluation results, the users can choose the most robust version via function calling instead of sticking to the newest version that is less robust. On the other hand, for platforms that do not provide the function calling service, we would still expect that the new model would be more robust than the old model. In sum, we will provide the specific context when we say `expected robustness` and especially discuss the application of function calling.
>
> [5].Zhou, Kun, et al. "Don't Make Your LLM an Evaluation Benchmark Cheater." arXiv preprint arXiv:2311.01964 (2023)
>
> [6]. https://twitter.com/keirp1/status/1724518513874739618

---

### Official Review · Reviewer_JE4Q · 2023-10-29

**Soundness:** 3 good
**Presentation:** 3 good
**Contribution:** 4 excellent
**Rating:** 8
**Confidence:** 4

**Summary:**

Prior studies have primarily centered on specific versions of the LLMs, neglecting the possibility of new attack vectors emerging for updated versions.
In this paper, the authors perform a thorough assessment of the robustness of the longitudinal versions of LLMs with a focus on GPT-3.5 and LLaMA. Their findings indicate that the updated model does not exhibit heightened robustness against the proposed adversarial queries compared to its predecessor.

**Strengths:**

* The research problem addressed in this paper is novel and previously underexplored. The findings are interesting and indicate that the majority of newly released LLMs lack robustness considerations. To promote responsible AI, technology giants should take into account the deployment of effective robustness-enhancing techniques and perform strict evaluations before releasing their latest LLMs. In particular, this paper demonstrates that both GPT-3.5 and LLaMA exhibit vulnerability to adversarial queries persistently across different versions.

* The authors employ diverse evaluation metrics to offer a comprehensive assessment of various model versions. They find that the performances of the LLMs need to improve as versions evolve steadily. Specifically, GPT-3.5 v0613 exhibits a discernible decline in performance in some specific tasks.

* This study involves a substantial workload. It encompasses the use of six distinct surrogate models and employs ten different settings for adversarial queries to ensure the thoroughness of the assessment.

**Weaknesses:**

* In this work, the authors primarily focus on assessing adversarial robustness exclusively within various iterations of LLMs. The authors should broaden the scope of their investigation to encompass additional thematic categories across diverse subject matter domains.
Furthermore, the authors should expand their evaluation efforts to encompass various dimensions of the model iterations they are not considering. Specifically, regarding the LLaMA model family, which includes models of varying architectural sizes, the authors should investigate and provide insights into the robustness of these models in the context of different parameter sizes.

* The motivation of this paper is clear. However, the authors should elaborate more on the process of generating adversarial examples by different surrogate language models

* I didn't find any results of LLaMA-30B, but the authors list this model in Section 4.2. I think the authors should provide some details about that.

**Questions:**

* Could the authors provide an explanation for the underlying reason that caused the LLMs not to exhibit heightened robustness over time? Additionally, could they discuss potential strategies to address this issue?

* Have the authors shared their findings with OpenAI or Meta to report this issue?

---

> ### Author Response · Authors · 2023-11-15
>
> Thanks for your constructive comments and suggestions; they are exceedingly helpful in improving our paper.
>
> >**The authors should investigate and provide insights into the robustness of these models in the context of different parameter sizes.**
>
> Thanks for the valuable suggestions. Our current research is focused on the `time` dimension since we consider it the most straightforward dimension for comparing the robustness of two closely related models. We would like to consider the suggested extensions for future work, i.e., `more subject matter domains`, `dimensions`, and `model parameter settings`.
>
> >**The authors should elaborate more on the process of generating adversarial examples by different surrogate language models.**
>
> We have added more details on how to generate the adversarial samples by different surrogate language models in Appendix.
>
> >**Results of LLaMA-30B**
>
> This is indeed a typo, and we have removed it from the paper.
>
> > **Could the authors provide an explanation for the underlying reason that caused the LLMs not to exhibit heightened robustness over time? Additionally, could they discuss potential strategies to address this issue?**
>
> The training details of LLMs are generally not available to the public. However, a simple observation is that the model trainer tends to consider model accuracy over robustness, partially because achieving adversarial robustness requires considerable additional effort. We have added the content to suggest the model trainer to use enhancing adversarial robustness methods in model upgrades in our discussion.
>
> > **Have the authors shared their findings with OpenAI or Meta to report this issue?**
>
> We have not done it yet, and we would like to do it once the review process is finished.

---

### Official Review · Reviewer_9mwd · 2023-10-31

**Soundness:** 3 good
**Presentation:** 3 good
**Contribution:** 3 good
**Rating:** 8
**Confidence:** 4

**Summary:**

This paper concludes that considerations for improving robustness should be integral when updating LLMs. This paper is well-written, with thorough experimental design and argumentation. It provides insightful contributions to the research on LLMs, emphasizing the importance of accounting for model version updates. Additionally, the proposed systematic design of adversarial queries should be considered a vital metric for assessing LLMs' performance.

**Strengths:**

1. **Variation in Robustness Across Model Versions.** This paper, for the first time, investigates the variability in the robustness of LLMs using model versions as a variable and adversarial samples as input objects. Meanwhile, the study encompasses 12 different versions of two of the most popular LLMs, i.e., ChatGPT and LLaMA.

2. **Comprehensive Analysis of Adversarial Attacks.** To comprehensively evaluate potential adversarial attacks on LLMs, this paper discusses 10 types of malicious queries (including zero-shot in-context learning and few-shot in-context learning scenarios). It also addresses different query elements, including descriptions, demonstrations, and questions by using multiple datasets such as PromptBench, GLUE, and AdvGLUE.

3. **Impact of Model Version Updates.** Experimental results demonstrate that updates in model versions do not significantly improve benign performance on various downstream tasks (e.g., results of CTS in Figures 3 and 4). Simultaneously, the robustness of the models shows a decreasing trend (as observed in Figures 3 and 4 for PDR results).

**Weaknesses:**

1. **Enhancing Adversarial Robustness in Model Upgrades.** The author(s) should add a discussion on strategies for improving adversarial robustness during version upgrades of LLMs. In fact, in my opinion, this is an important part that can inspire the community to proceed further research on safety and security of LLM.


2. **Considering New Features in Model Evolution.** Note that model updates may introduce new features. For example, recent versions of ChatGPT allow internet access. Future research could explore the correlation between online connectivity and robustness.

**Questions:**

None

---

> ### Author Response · Authors · 2023-11-15
>
> Thank you for your valuable feedback and pointing out the many positive aspects of our work. Below, we’ll address the negative aspects you mentioned.
>
> >**Enhancing Adversarial Robustness in Model Upgrades.**
>
> This is a good point. We have discussed the strategies in the paper to suggest the model trainer to use enhancing adversarial robustness methods in model upgrades.
>
> >**Considering New Features in Model Evolution.**
>
> This is a promising direction for our future work. We have added the content about the new feature of GPT-3.5 in the paper. For the browse with bing of ChatGPT, it may be expected that new features are also vulnerable to existing attacks and may also introduce new attack surfaces that make LLMs even more vulnerable.

---

### Official Review · Reviewer_4jLb · 2023-11-01

**Soundness:** 3 good
**Presentation:** 3 good
**Contribution:** 3 good
**Rating:** 6
**Confidence:** 2

**Summary:**

Large language models (LLMs) have significantly improved many cross-domain tasks. However, these models often overlook the impact of security and privacy when upgrading, which can lead to unintended vulnerabilities or biases. Previous studies have predominantly focused on specific versions of the models and disregard the potential emergence of new attack vectors targeting the updated versions. This paper conducts a comprehensive assessment of the robustness of successive versions of LLMs, vis-`a-vis GPT-3.5 and LLaMA.

**Strengths:**

- Well-written.
- The experiment was comprehensive.

**Weaknesses:**

- Hope the author can provide a more detailed description of "zero-shot ICL learning" and "few-shot ICL learning" in Figure 2.
- What does the second column "Adversarial Query" in Table 1 mean? Please clarify.
- "For instance, on the SST2 dataset, when applying BertAttack (Li et al., 2020) to create the adversarial description, the Robust Test Scores (see Section 4.3) for both versions of GPT-3.5 are almost 0." , which table or figure is being described specifically? Please clarify.

**Questions:**

Please see "Weaknesses".

---

> ### Author Response · Authors · 2023-11-15
>
> Thanks for your encouraging words and constructive comments. We sincerely appreciate your time reading the paper, and our point-to-point responses to your comments are below.
>
> >**Hope the author can provide a more detailed description of "zero-shot ICL learning" and "few-shot ICL learning" in Figure 2.**
>
> If the reviewer means the description of the two terms, we will further clarify that we follow [1,2,3] on this terminology. Zero-shot learning means that the query includes only the description and the question but **without any demonstrations**, while few-shot learning means that the query also includes a few `demonstrations`. We have added the explanation in the caption of Figure 2. If the reviewer means the `description` component in the query, please see the detailed definitions in Section 2.1.
>
> [1]. Yongqin Xian et al. Zero-Shot Learning - A Comprehensive Evaluation of the Good, the Bad and the Ugly. IEEE Transactions on Pattern Analysis and Machine Intelligence, 2019.
>
> [2]. Tom B. Brown et al. Language Models are Few-Shot Learners. In Annual Conference on Neural Information Processing Systems (NeurIPS). NeurIPS, 2020.
>
> [3]. Jason Wei et al. Finetuned Language Models are Zero-Shot Learners. In International Conference on Learning Representations (ICLR), 2022.
>
> >**What does the second column "Adversarial Query" in Table 1 mean? Please clarify.**
>
> `Adversarial Query` refers to the query that contains the adversarial content in any of its three components (`description`, `question`, and `demonstrations`), as defined in *Eq. (1)* and *Eq. (2)*. Specifically, each component can be clean (denoted as C) or adversarial (denoted as A).
> Therefore, in Table 1, the adversarial query `AC` means a zero-shot learning-based query that consists of an adversarial `description` and a clean `question`, and `AAC` means a few-shot learning-based query that consists of an adversarial `description`, an adversarial `question`, and clean `demonstrations`. We will describe it more clearly in Section 5.1.
>
> >**"For instance, on the SST2 dataset, when applying BertAttack (Li et al., 2020) to create the adversarial description, the Robust Test Scores (see Section 4.3) for both versions of GPT-3.5 are almost 0." Which table or figure is being described specifically? Please clarify.**
>
> Sorry, we indeed made a mistake here. It should be ''For instance, on the SST2 dataset, the average result of Robust Test Scores of zero-shot learning for both versions of GPT-3.5 dropped from 85.093% and 87.390% to 37.210% and 20.652%, respectively (see Figure 3).''

---

### Meta-Review · Area_Chair_z1io · 2023-12-08

**Metareview:**

This paper studies how adversarial robustness of LLMs (for in-context learning) evolves over time as LLMs are updated and upgraded.
While this question is undoubtably interesting, it is unclear what conclusions to draw from the current paper's experiments.
The only experiment that actually studies a model version change is the one on ChatGPT 3.5, but this is for a single change.
For LLama models, we have a change of model (from Llama to Llama2) and a change of fine-tuning (from Llama2 to Llama2-chat).

Of course we should expect there to be some change in performance when we change the model. The paper unsurprisingly finds that this is the case, but the magnitude of the changes is small, and often not statistically significant. E.g., the paper notes that ChatGPT's update leads to a significant drop on MNLI, and yet the error bars seem to overlap.
For LLama models we do a see a more significant drop on many benchmarks in the few-shot learning setting, but this is true even on clean data. So this is likely due to the chat alignment more than anything else.

In summary, while I find the question raised by this paper interesting, it is not clear to me that the paper's experiments allow us to draw any conclusions about how robustness evolves over time.

**Justification For Why Not Higher Score:**

The reviewers seemed to mostly like the paper, but didn't comment on the statistical significance of the presented results, which is unclear.
I think the research question is interesting and worth studying more deeply, but the current paper conflates a number of model changes and does not give convincing evidence of how robustness evolves over time.

**Justification For Why Not Lower Score:**

N/A

---

### Decision · Program_Chairs · 2024-01-16

Reject